# Repetitive and Inflexible Active Coping and Addiction-like Neuroplasticity in Stressed Mice of a Helplessness–Resistant Inbred Strain

**DOI:** 10.3390/bs11120174

**Published:** 2021-12-10

**Authors:** Simona Cabib, Paolo Campus, Emanuele Claudio Latagliata, Cristina Orsini, Valeria Tarmati

**Affiliations:** 1Department of Experimental Neurosciences, IRCCS Fondazione Santa Lucia, 00179 Roma, Italy; claudio.latagliata@gmail.com; 2Department of Psychology, Sapienza University of Rome, 00185 Roma, Italy; cristina.orsini@uniroma1.it (C.O.); valeria.tarmati@uniroma1.it (V.T.); 3Department of Psychiatry, University of Michigan, Ann Arbor, MI 48109, USA; pcampus@umich.edu

**Keywords:** accumbens, anxiety, avoidance, behavioral sensitization, D2 dopamine receptors, frontal cortex, genotype, mesencephalon, motivation, reversal learning, stereotypy, ventral striatum

## Abstract

Dysfunctional coping styles are involved in the development, persistence, and relapse of psychiatric diseases. Passive coping with stress challenges (helplessness) is most commonly used in animal models of dysfunctional coping, although active coping strategies are associated with generalized anxiety disorder, social anxiety disorder, panic, and phobias as well as obsessive-compulsive and post-traumatic stress disorder. This paper analyzes the development of dysfunctional active coping strategies of mice of the helplessness–resistant DBA/2J (D2) inbred strain, submitted to temporary reduction in food availability in an uncontrollable and unavoidable condition. The results indicate that food-restricted D2 mice developed a stereotyped form of food anticipatory activity and dysfunctional reactive coping in novel aversive contexts and acquired inflexible and perseverant escape strategies in novel stressful situations. The evaluation of FosB/DeltaFosB immunostaining in different brain areas of food-restricted D2 mice revealed a pattern of expression typically associated with behavioral sensitization to addictive drugs and compulsivity. These results support the conclusion that an active coping style represents an endophenotype of mental disturbances characterized by perseverant and inflexible behavior.

## 1. Introduction

Clinical findings point to a major role of dysfunctional coping with life adversities (stressors) in the development, persistence, and relapse of mental diseases [1,2,3,4]. Therefore, the psychobiological processes responsible for the development of dysfunctional coping styles are relevant targets of preclinical studies in animal models.

The term coping refers to behavioral and intrapsychic attempts to master, minimize, or tolerate stressful experiences. These responses are activated by the appraisal of an experience as stressful, i.e., overwhelming, to the subject [5,6,7,8]. There are different coping strategies, which can be adaptive or dysfunctional depending on the stressor. Active and proactive coping strategies are effective when there are ways to solve the stressful problems, or to escape or avoid them, but whenever stressors cannot be controlled, eliminated, or escaped, passive or emotion-focused strategies are required to mitigate their emotional and physiological impact [7,9,10,11,12].

In recent years, the development of animal models of dysfunctional stress coping has been favored by a growing interest in the mechanisms supporting resilience to traumatic stress [13,14,15], but most of these models focus on phenotypes associated with passive coping. In the absence of a clear and shared definition of dysfunctional coping, a diffuse view identifies passive strategies as less functional than active ones or even as straightforward models of depressive symptoms. However, the idea that passive coping is a dysfunctional coping strategy has been challenged by many reviews of clinical and preclinical data [10,11,16], and active avoidance strategies are overexpressed in generalized anxiety disorder, social anxiety disorder, panic, and phobias, as well as in obsessive-compulsive and post-traumatic stress disorders [16]. Additionally, individual coping strategies are influenced by previous experiences and genotype; thus, dysfunctional coping could be the outcome of a diathesis–stress mechanism [2,4,6,12,17,18].

In the present study, we tested the hypothesis that mice of the DBA/2J (D2) inbred strain exposed to a temporary reduction in food availability develop dysfunctional active coping. Converging behavioral and neurobiological evidence indicates a bias toward the use of active coping strategies to deal with uncontrollable or unavoidable stress [15], and a genotype-specific neurobiological adaptation to chronic or repeated stress [12] in D2 mice. Moreover, persistent unsuccessful foraging is a severe chronic uncontrollable and unavoidable stressful experience in nature. In the laboratory, food restriction protocols reproduce this condition when animals are not trained in food-reinforced tasks. These protocols require removing food from the home cage and only making it available within a specific time window. Coping with the absence of a food reserve requires foraging, and food-reinforced training is a form of food searching activity. Because food is generally made available in the home cages after training sessions, trained animals experience successful foraging. Instead, food-restricted rodents that are not trained in food-reinforced tasks and have running wheels available lose more body mass than rodents without this opportunity, and their behavior is considered a model of anorexia [19,20]. Food-restricted rodents that are not allowed to run the wheel develop behavioral sensitization to addictive drugs [21,22,23,24].

The first experiment of this study tested food anticipatory activity (FAA) as a measure of foraging motivation. FAA has been associated with foraging [25,26,27], which, as discussed, is a form of active coping; it was shown that food motivation determines the intensity of expressed FAA, though not the timing [28]. The experiment compared the behavior expressed by mice of the D2 strain with that expressed by mice of the genetically unrelated C57BL/6 inbred strain (B6). Although FAA is usually measured by wheel running, this approach could not be employed in our experiments because mice of the D2 inbred strain with unrestricted access to wheels do not suspend running to feed when food is available and starve to death [19]. Therefore, we measured home cage behaviors expressed during the 30 min preceding food delivery.

A second set of experiments was used to evaluate the effects of previous restricted feeding on behavior expressed by B6 and D2 mice in two standard anxiety tests: the elevated plus maze (EPM) and the elevated T maze (ETM). Active coping is modeled in rodents by avoidance or escape of stressful situations, and the two tests measure avoidance and escape of dangerous areas.

A final behavioral experiment was used to evaluate the effects of previous restricted feeding on the flexibility of an acquired escape response. As different coping strategies are effective in different stressful situations, inflexible and perseverant coping strategies are considered dysfunctional [29,30]. Thus, mice of the D2 strain were tested for reversal learning, which is a measure of behavioral flexibility [31], of an escape strategy acquired in an escapable stressful situation: the submerged T-maze.

Additional experiments were used to evaluate the accumulation of DeltaFosB fostered by the experience of restricted feeding in different brain areas of mice of the D2 and B6 inbred strains. The transcription factor DeltaFosB is a highly stable splice product of the immediate early gene FosB that slowly accumulates during repeated and chronic stress experiences [32,33], including food [34] and caloric restriction [35]. Moreover, DeltaFosB expression within the brain reward system is fostered by chronic administration of antidepressants and it was reported to prevent the expression of passive coping under acute stress challenge [36]. Finally, the transcription factor also accumulates in the brain reward system in response to prolonged experience with addictive drugs; it was proposed as a marker of addiction-associated neuroplasticity [37].

## 2. Materials and Methods

### 2.1. Animals and Housing

Male mice of the inbred DBA/2J (D2) and C57BL/6J (B6) strains (Charles River, Como, Italy) were purchased at 6 weeks of age and housed in groups of four in standard breeding cages with food and water at libitum on a 12 h dark to light cycle (lights on between 07:00 a.m. and 7:00 p.m.) at a temperature of 22 ± 1 °C. These two inbred strains are genetically unrelated and are the parental strains of a panel of recombinant inbred strains most used for genetic studies. Thus, the literature offers a useful wealth of data on phenotypic differences between the mice from the two strains.

When mice reached 7 weeks of age, they were assigned to different experimental groups: free fed (FF) mice were individually housed and given food once daily in a quantity adjusted to exceed daily consumption (15 g, on the bottom of the cage) for 14 consecutive days; no food (NF) mice were individually housed and given food once daily in a quantity adjusted to foster a 25% weight loss in the first 3 days and to keep the weight of each mouse stable (at 75% initial weight) for the following 9 days of the differential housing, and NF+48 mice were food-restricted (same protocol described for NF mice) for 12 consecutive days and then given food ad libitum for 48 h before testing. The NF+48 protocol allows animals to recover from the physical strain of restricted feeding that could influence behavioral responses expressed in challenging testing situations such as the water T-maze. Therefore, we used mice from the NF+48 group for all behavioral experiments. NF mice were used to test the effects of restricted feeding on the brain FosB/ DeltaFosB immunostaining because although DeltaFosB is stable and slowly accumulates over time in chronic protocols, FosB expression can be acutely and temporarily elicited by changes in environmental conditions.

All mice were weighed once every two days and received food once daily one hour before the light–dark shift to reach and maintain 85% of the mouse’s initial weight. All experimental manipulations were performed in the 2nd half of the light period and terminated 15 min before daily food delivery. Food-restricted mice were provided food in a quantity adjusted to foster a loss of 15% of initial weight within the first 3 days of individual housing and to maintain this weight for the following 9 days.

Experiments were conducted according to the Italian National Law on the Use of Animals for Research (DL 116/92) in line with EU Directive 2010/63/EU for animal experiments.

### 2.2. Food Anticipatory Activity

In this study, we measured home cage behaviors expressed by FF and NF mice of the D2 and B6 strains during the 30 min that preceded daily food delivery. The presence of 8 different behavioral items: still (STILL), sitting with little or no movement; locomotion (LOC), horizontal activity; rearing (REAR), exploration of the cage walls; climbing (CLIMB), clinging with four paws to the cage cover; grooming (GROOM), head or body grooming; digging (DIG), digging into the sawdust; chewing (CHEW), either sawdust or food pellets; and drinking (DRINK) was recorded (0–1) each 2 min by a trained observer unaware of the feeding condition. Total scores (frequency) of each behavioral item collected over 30 min were used as the dependent variables. Statistical analyses of data collected on the last day of restricted feeding (day 12) were conducted by 2-way ANOVAs for independent measures (strain: B6, D2; and feeding condition: FF, NF). All statistical analyses reported in this paper were conducted on Prism GraphPad. The CLIMB expressed by NF mice of the two strains in the course of the 14 days of differential housing was statistically analyzed by 2-way ANOVA for repeated measures (Strain: B6, D2; days of differential housing: 1, 3, 7, 9, 12, +48 h) as well as the simple effects of the factor days within each strain.

### 2.3. Elevated Plus Maze (EPM) and Elevated T Maze (ETM)

The EPM used for these experiments was a gray Plexiglas apparatus with two open arms (27 × 5 cm) and two enclosed arms (27 × 5 × 15 cm). Arms extended from a central platform (5 × 5 cm) elevated to a height of 38.5 cm above floor level by a central pedestal. A slightly raised edge (0.25 cm) was placed around the open arms’ perimeter. For ETM testing, the entrance to one of the two enclosed arms was blocked by a Plexiglas door of the same color and texture as the apparatus.

All behavioral experiments were conducted in a sound-attenuated cubicle in which a 60 W lamp (1.50 m above the apparatus) was the only source of illumination. All tests were videotaped by means of a camera placed within the experimental cubicle and linked to a monitor and VCR placed in an adjacent room. In all cases, animals were transported to the cubicle within their home cages and left undisturbed for at least two hours before the test.

Independent groups of mice from each strain were tested using the EPM and ETM. In the EPM, individual mice from two experimental groups (FF, NF+48; *n* = 7–8, for each strain) were gently placed on the central platform of the maze facing an open arm and left to explore the maze for the subsequent 5 min. Frequency of entries and time (s) spent in the arms were taken as behavioral measures. Statistical analyses (2-way ANOVAs with strain B6 or D2 and the four housing conditions as independent variables) were performed on total (open + closed) arm entries, percentage of entries into the open arms, and percent of time spent in the open arms.

In the ETM, individual mice (FF, NF+48; *n* = 12 each) were tested for each strain. Each mouse was placed facing the end of the closed arm. The time required to leave the arm (with all four paws) was recorded (baseline latency). Mice that reached the cut-off limit (180 s) in the baseline session (AV 1) were gently pushed to enter an open arm. Mice that entered the open arms were removed from the maze and left undisturbed in their home cage for 60 s. The procedure was repeated in two subsequent trials (avoidance 2 (AV2) and avoidance 3 (AV 3)). After the last avoidance session, the mouse was placed at the end of one open arm (with left and right balanced within each group), and the time required to leave the arm (i.e., to enter the enclosed arm with the four paws) was recorded (escape latency).

Latency to enter the open arms of the ETM was evaluated by a 3-way mixed model ANOVA with two independent variables: strain (B6 or D2) and housing condition (FF or NF+48), and a repeated measure: avoidance sessions (AV1, AV2, and AV3). Significant interactions between housing and sessions were also analyzed post hoc within each strain. Finally, escape latencies (latency to leave the open arm on the forced trial) were statistically evaluated by a 2-way ANOVA for independent variables (strain: two levels = B6 or D2; housing two levels = FF or NF+48).

### 2.4. Training and Reversal Testing in the Water T-Maze (WTM)

In the present study, we employed the WTM to test reversal learning following restricted feeding because we were interested in how NF D2 mice adapt a successful active coping strategy to variable contingencies. D2 mice are very poor spatial learners; thus, it would have been difficult, if not impossible, to use a Morris water maze [38,39]. Instead, the T-maze is a classic double-solution test; it does not require spatial learning, and we previously showed the rapid acquisition and extinction of the escape strategy by NF+48 D2 mice in the water version of this test [40].

A total of 24 D2 mice were used for the WTM experiment (12 per feeding condition: FF and NF+48). The WTM apparatus consisted of a clear plexiglass T-maze inserted within a dark circular water maze (100 cm diameter, 35 cm height). Each arm measured 5 cm in width and 40 cm in length, with 30 cm high walls. The maze was filled with water (22 ± 2 °C), and non-toxic white paint was added to ensure opacity. A moveable clear plexiglass escape platform (14 × 14 cm) was located at the left or right end of the maze arm and submerged 2 cm from the surface of the water. Experiments involved 5 consecutive days of training in which mice were placed in the start arm (south) of the maze and then allowed to swim to find an invisible escape platform located at the end of the west or east arm of the maze (balanced within each group). Mice remained on the escape platform for 10 seconds before being transferred to a holding cage for a 60 s inter-trial interval. Each of the 5 daily training sessions consisted of 6 consecutive trials that had a maximum duration of 60 s. Mice able to successfully complete 5 training trials on the 3 final training sessions (errors ≤ 2) were moved into the reversal session, consisting of the same protocol adopted for training, with the exception that the platform was placed in the opposite arm relative to training. A single session was used because we were interested in resistance to abandoning the previously acquired strategy (perseveration) rather in the ability to learn the new rule [31].

A digital video camera located above the apparatus was connected to a computer in the same room and used to track swimming paths using EthoVision software (Noldus, The Netherlands). Mean latency to escape (s), distance swum (cm), and mean number of entries in the reinforced arm (correct trials) across the 6 daily trials were automatically computed. Statistical analyses of the 3 behavioral measures were performed by two-way ANOVAs (independent factor: differential housing FF, NF+48: within factor: training trials 1–5 or the 5th session of training vs. reversal test.

### 2.5. Quantification of FosB/DeltaFosB Immunoreactivity

FosB and DeltaFosB immunoreactivity was examined in FF and NF mice from each strain (*n* = 6 per group). All mice were killed by decapitation on the 12th day of differential housing within the first half of the light period. These mice were not used for any other experimental purpose. After removal, brains were immersed in chilled 10% neutral buffered formalin, stored overnight, and then cryoprotected in 30% sucrose solution at 4 °C for 48 h [41]. Frozen coronal sections (40 µm thickness) were cut through the whole brain with a sliding microtome and then immunolabelled with the immunoperoxidase method as previously described [41,42]. FosB ⁄DeltaFosB antigens were detected with a rabbit polyclonal antiserum (sc-48, 1:10,000, Santa Cruz Biotechnology) raised against an internal region of FosB protein that is also present in DeltaFosB isoforms (1:1000 sc-48, Santa Cruz Biotechnology, Santa Cruz, CA, USA). Peroxidase labeling was obtained by the standard avidin–biotin procedure (Vectastain ABC elite kit, Vector Laboratories, diluted 1:500), and the chromogenic reaction was developed by incubating sections with metal-enhanced DAB (Vector Laboratories). Part of the tissue samples from the NAc Sh was treated for c-fos immunolabeling using rabbit anti-c-fos (1/20,000; Oncogene Sciences Cambridge, MA, USA) as the primary antibody.

Sections were analyzed using a Nikon Eclipse 80i microscope equipped with a Nikon DS-5M CCD camera as previously described [42]. Specimens were subjected to quantitative image analysis using the public domain image analysis software IMAGEJ 1.38 g for Linux (W.S. Rasband, ImageJ, U.S. National Institutes of Health, Bethesda, Washington, DC, USA) [41,42] For quantification, the striatum was divided into four compartments: dorsomedial (DMS), dorsolateral (DLS), nucleus accumbens core (NAc Co), and shell (NAc Sh); the hippocampus into CA1, CA3, and dentate gyrus (DGY); the amygdala in lateral (LA), basal (BaA), and central (CeA); and the mesencephalon in ventral tegmental area (VTA); substantia nigra pars compacta (SNc); and substantia nigra pars reticulata (SNr). Immunoreactive nuclei density was measured and is expressed as number of nuclei/0.1 mm^2^.

Immunostaining of tissue samples from the two strains was performed in different batches; the density of the immunostained nuclei for each area was calculated. We initially tested the hemisphere-dependent effects of restricted feeding (2-way ANOVA for repeated measure: left and right hemisphere, with feeding experience: FF, NF, as independent factors) because in previous studies we observed a lateralized effect of the NF experience on the left hemisphere of D2 mice [42,43], but only found a significant interaction for SNr. Thus, we considered FF and NF data from other sampled area by *t*-test (two-tailed). In addition, we evaluated the relationship between FosB/deltaFosB and c-fos expression in the NAc Sh by linear regression to weigh the effects of the experimental manipulations on the expression of acutely inducible transcription factors. The results did not reveal any significant correlation between the two sets of data.

## 3. Results

### 3.1. Anticipatory Activity

Table 1 presents the data on the home cage behavior expressed in the 30 min preceding food delivery on the 12th day of differential housing. Statistical analyses only revealed a significant interaction between the factors strain and housing for STILL (F (1,24) = 5.05; *p* < 0.05) and CLIMB (F (1,24) = 27.45; *p* < 0.0001) behaviors. Post hoc comparison (Tukey’s correction) revealed a significant reduction in inactivity in NF mice of the two strains, although the effects were more pronounced in D2 mice (Table 1), and an increase in CLIMB expressed by mice of both strains, although the effect of NF on this behavior was significantly higher in mice of the D2 strain.

The analysis of CLIMB behavior over the 14 days of differential housing (Figure 1) revealed a significant interaction between the factors strain and day of data collection: F (5,60) = 27.97; *p* < 0.001). Post hoc data analysis showed a highly significant increase in climbing on days 7, 9, and 12 of restricted feeding in D2-strain mice. As for B6 mice, a significant increase in climbing was only observable on the ninth day of restricted feeding. Climbing expressed following 48 h of free access to food was no different from that expressed on the first day of differential housing.

### 3.2. Performance in EPM and ETM

Data collected in the EPM are presented in the top of Figure 2. Statistical analyses performed by two-way ANOVA revealed a significant interaction between strain and housing for total entries (open and closed arms): F (1,28) = 9.217; *p* < 0.01; number of entries in the open arms: F (1,28) = 10.15; *p* < 0.01; and time spent in open arms (seconds): F (1,28) = 16.39; *p* < 0.005. FF mice of the D2 strain spent less time in the open arms than B6 mice, and the experience of restricted feeding eliminated this strain difference by increasing time spent in the open arms by D2 mice.

Behavioral data collected in the ETM are shown in the bottom of Figure 2. The three-way ANOVA on latency to leave the protected arm revealed a significant interaction between factors session and feeding condition (F (2,88) = 3.508; *p* < 0.05) and a highly significant interaction between factors session and strain (F (2,88) = 21.62; *p* < 0.001). The interaction between session and feeding factors tested for each strain by two-way ANOVA was only significant for data collected from D2 mice (F (2,44) = 3.895; *p* < 0.05) because of a progressive decrease in the latency to leave the protected arm in food-restricted mice of this inbred strain. Both FF and NF+48 mice of the B6 strain showed a progressive increase in the latency to emerge from the protected arms (main effect of the factor session: F (2,44) = 17.43; *p* < 0.001) from AV1 to AV3.

Finally, a significant interaction between strain and housing was found for latency to escape from open arms (F (1,44) = 4.302; *p* < 0.05). FF mice of the D2 strain showed longer latency to escape from the open arms than FF B6 mice. A sharp decrease in the escape latency by D2 mice eliminated the strain difference in NF+48 mice (Figure 2, bottom right).

### 3.3. Performance in WTM

Data collected from the WTM are presented in Figure 3. Statistical analyses of the number of correctly performed trials, the mean latency to enter the reinforced arm, and the total distance swum on each session by two-way ANOVA only revealed a significant main effect of the training sessions for all parameters (F (4,88) = 9.886; *p* < 0.0001, F (4,88) = 22.95; *p* < 0.0001, F (4,88) = 13.49; *p* < 0.0001, respectively). Post hoc evaluation of differences with performance on day 1 performed on the main effect of training sessions revealed that latency and distance measures decreased over time as the number of correct trials increased. Moreover, performance stabilized between the third and the fifth training sessions (Figure 3, right).

A total of nine mice from each experimental group showed at least 80% correct responses over the three final days of training and were selected to test reversal learning. Figure 3 (right) shows the data collected on the fifth training session and on the single reversal session by the mice selected from the two experimental groups. The two-way ANOVA revealed a significant interaction between the factors of session and feeding condition for the three parameters (correct trials: F (1,16) = 5.388; *p* < 0.05, mean latency: F (1,16) = 4.531; *p* < 0.05; total distance: F (1,16) = 12.38; *p* < 0.005). However, a significant interaction between feeding condition and reversal experience was found for correct trials due to a larger reduction in the performance by NF+48 mice (Figure 3, right).

### 3.4. Immunohistochemical Results

Figure 4 shows samples of FosB/DeltaFosB immunostaining within the striatal complex. Figure 5 reports data (mean density ± SEM) on FosB/DeltaFosB immunostaining in the sampled brain areas of FF and NF mice from each strain. A lateralized effect of restricted feeding was only found in the SNr of D2 mice (significant interaction hemisphere × feeding: F(1,10) = 7.389; *p* < 0.05) because of a significant overexpression of FosB/DeltaFosB immunostaining in the left hemisphere of NF mice (Figure 5, bottom insert). The experience of restricted feeding enhanced FosB/DeltaFosB immunostaining in the SNc (B6: t(10) = 2.892; D2: t(10) = 2.476) and VTA (B6: t(10) = 3.266; D2: t(10) = 2.509) of mice from both inbred strains; only NF B6 mice showed enhanced expression in the BaA (t(10) = 2.257), DLS (t(10) = 3.514), and SNr (t(10) = 2.878); and only NF D2 mice showed enhanced expression in the hippocampus CA1t(10) = 4.032) and NAc Sh t(10) = 6.714).

## 4. Discussion

The collected experimental evidence offers support to the hypothesis that restricted feeding fosters development of a perseverant and inflexible reactive coping style by D2 mice as well as of the neuroadaptation of the mesostriatal system associated with perseverant and inflexible behavior.

### 4.1. Stereotyped Active Coping with the Food Restriction Protocol by Mice of the D2 Strain

As discussed in the Introduction, we used FAA as a measure of motivation because the intensity of this phenotype is dependent on the strength of food motivation [28]. In line with this view, NF mice from both strains developed FAA (significant reduction in STILL behavior in the 30 min preceding food deliver (Table 1), and cage cover climbing (CLIMB) disappeared in mice of both strains given food ad libitum for 48 h, indicating that food motivation is a requirement for the expression of this behavior (Figure 1). Most D2 mice expressed maximal levels of CLIMB on days 9 and 12 of restricted feeding and near maximal levels on day 7 (Figure 1), whereas B6 mice only exhibited a temporary moderate increase in this behavior on day 9 (Figure 1). Because fluctuation in food motivation by NF B6 mice during restricted feeding would be unlikely, other mechanisms besides motivation are reasonably involved in the strain difference for this behavioral phenotype.

In standard living conditions, food is located over the cage cover; therefore, climbing is an effective food searching strategy. Climbing is a dynamic behavior: mice jump to reach the metal grid covering the cage and hang there with their four paws to eat when food is available. When food is removed, they jump up, briefly explore the grid, then jump back to the ground. CLIMB observable in NF mice of the D2 strain can be a constant jumping up and down from the grid. However, in both FF and NF conditions, food is placed on the cage floor. Thus, digging is the typical FAA observable in FF D2 mice and in both FF and NF B6 mice (Table 1). The progressive increase in an unreinforced food searching strategy by NF D2 mice (Figure 1) can indicate development of a stereotypy: an inflexible and perseverant behavior with no apparent function that is observable in animals exposed to suboptimal environmental conditions [44]. In line with this view, stereotypies expressed by animals within their home cages are developed from frustrated motivation [45]. Moreover, climbing behavior is induced by administration of the dopamine (DA) direct agonist apomorphine [46], a drug known to induce stereotypic behavior in rodents [47,48]. Finally, the behavioral effects of apomorphine are strain-specifically enhanced in NF+48 D2 mice [49].

Finally, the perseverant form of cage-cover climbing developed by NF D2 mice is most probably dependent on the reduction in striatal DA D2 receptors induced by restricted feeding in mice of this inbred strain [42]. The low striatal availability of these receptors is consistently associated with perseverant and compulsive behavior [50,51,52,53,54], and a recent study showed that overexpression of striatal D2 receptors reduces expression of FAA by food-restricted mice [28].

### 4.2. Dysfunctional Reactive Coping with Novel Aversive Situations by Food-Restricted D2 Mice

Rodents are nocturnal animals that live in borrows and are predated by rapacious birds. Thus, they avoid open unprotected spaces and try to escape when they find themselves within such environments. The open arms of an elevated maze are the most unprotected (they are open to attacks from four sides). Thus, when tested in EPM and ETM, experimental animals explore a novel environment characterized by a safe section (the protected arms) and a dangerous section (the unprotected arms). To be confined in a novel, potentially dangerous, context is a moderate stressor that can be coped with by learning to avoid the open arms. Thus, rodents are expected to explore the apparatus and develop avoidance for the unprotected arms. EPM and ETM are used to measure anxiety levels, and different behaviors can be used to this aim; however, in these experiments, we were interested in measuring open arm avoidance.

EPM experiments revealed a strain difference in behavior expressed by FF mice. FF mice of the D2 strain made fewer entries into and spent less time within the open arms than B6 mice of the same experimental groups (Figure 2 top), in line with previous findings collected by different laboratories [55,56]. A total of twelve days of restricted feeding followed by 48 h of free access to food (NF+48) increased both entries and time spent in open arms by D2 mice only. As open arms avoidance in EPM is reduced by anxiolytics, the behavioral response exhibited by NF+48 D2 mice could indicate reduced anxiety. On the other hand, mice of the D2 inbred strain show high levels of freezing in novel environments [57,58], and increased exploration in various testing conditions was observed in mice of this inbred strain following experience of the chronic mild stress protocol (CMS) [58]. Thus, a shift from passive to active avoidance of the unprotected arms could explain the behavior expressed by D2 mice following restricted feeding. To test this hypothesis, we turned to the ETM test, which allows separate escape and passive avoidance of the open arms [59,60].

The behavior exhibited by FF B6 mice in the ETM (Figure 2, bottom: latency) was similar to that previously observed in rats, i.e., the latency to leave the protected arm increased with experience of the unprotected arms, indicating the acquisition of passive avoidance [59,60]. Moreover, FF B6 mice placed inside one of the unprotected arms at the end of this procedure (Figure 2, bottom: escape latency) immediately escaped to the protected one, showing the ability to express an active coping response appropriate to the context [59,60]. Instead, FF D2 mice showed high levels of freezing, which interfered with the exploration of both the protected and unprotected arms. Latencies to enter the unprotected arms on the first avoidance trial were similar to those observable in fully trained B6 mice (Figure 2, bottom: latency). Finally, when placed in the open arms following the last of the training sessions, FF D2 mice showed high escape latencies (Figure 2, bottom: escape latency), also related to an immediate freezing response. Freezing is a passive defensive reaction against possible impending dangers because it reduces the risk of being discovered by a predator. Therefore, behavior expressed by FF D2 mice during avoidance training indicates a passive coping response to the novel environment. Although this is the first report of strain-specific behavioral responses to ETM, a pattern of response similar to that observed in D2 mice was reported by a study in mice of an outbred albino strain [61].

As observed in EPM, restricted feeding only changed the behavioral response expressed by mice of the D2 strain. NF+48 D2 mice showed a progressive decrease in latency to escape from protected arms and immediate escape from the unprotected arms at the end of the avoidance training than FF mice. The latter response rules out the hypothesis of reduced fear of the open arms by NF+48 mice of the D2 strain. Notably, NF+48 D2 mice did not explore the open arms of the EPM: they darted into them and froze there or immediately ran back to the protected area of the maze. We did not expect this behavior and the registrations obtained did not allow its measurement. Nonetheless, the significant reduction in the latency to escape from the unprotected arms at the end of the passive avoidance training is in line with the behavior observed during passive avoidance training. Finally, the progressive reduction in the latency to dart into the unprotected arms by the end of the avoidance training indicates impaired acquisition of an avoidance response that is associated with species-typical survival needs.

The behavioral data collected in the two anxiety tests support the conclusion that D2 mice are characterized by maladaptive levels of anxiety in a novel environment because either extreme freezing or extreme reactive escape leads to impaired exploration and enhanced risk. Moreover, the previous experience of restricted feeding did not influence the emotional arousal elicited by the mildly stressful situation in mice of either strain, but it fostered the development an active coping response that further reduced the ability of these mice to adapt to the novel environment.

### 4.3. Food-Restricted D2 Mice Acquire an Active Coping Response as an Inflexible and Perseverant Strategy

In previous studies, we have observed that NF+48 D2 mice are capable of acquiring a passive coping strategy (immobility in an inescapable or unavoidable stressful situation), but relapse into active coping when retested in the situation, suggesting impaired retention of the learned coping response [40,62]. Here, we evaluated the effects of previous restricted feeding on acquisition and retention of an active coping response in an escapable stressful situation, the WTM. Regardless of the feeding condition, D2 mice acquired the correct escape strategy by the third of the five training sessions (Figure 3, left). We used a short training protocol to prevent development of a habitual response because habitual responses are resistant to changes in contingences and their development is fostered by overtraining and facilitated by stress [63].

On the reversal session performed 24 h after the end of training, NF+48 mice showed a more severe impairment in the performance than FF mice. There is evidence that errors observable in the first reversal session are due to perseveration into the no-more-successful rule, whereas performance on subsequent sessions is influenced by learning processes [31]. Therefore, the behavior expressed by NF+48 D2 mice on this single reversal session supports the hypothesis of the development of perseverant responses by these mice. Impaired reversal learning is widely used to reproduce the cognitive inflexibility associated with different neurological and psychiatric disorders in animal models [31]. Moreover, impaired reversal learning in WTM characterizes mouse models by repetitive and perseverant behavior [64]. Finally, WTM is a controllable and escapable stressful condition that can be successfully coped with by reactive coping strategies that are acquired to be rapidly implemented in similar situations [65]. Thus, present data offer strong support to the hypothesis that previously food-restricted D2 mice develop inflexible and perseverant active coping in novel stressful situations.

Impaired reversal learning has been observed in stressed humans and nonhuman animals [66,67,68], although facilitation and no effect of stress have also been observed in rodent models [68,69,70]. The discrepancy amongst findings is most probably dependent on differences in stress protocols. As an example, experience of acute stress is reported to facilitate reversal learning, whereas chronic or repeated stressors foster the opposite effect [68]. Moreover, in the present experiments, mice were trained to acquire the escape strategy following the end of the stressful experience (12 days NF+48 h of free feeding) and tested for reversal learning 24 h later, whereas, in studies showing opposite results, the stressful protocol was applied between acquisition of the instrumental response and the reversal training [69,70].

Finally, the inflexibility of the acquired escape response could depend on the abovementioned reduction in striatal D2 receptors fostered by restricted feeding in mice of the D2 strain. Mice with DA signaling restricted to the dorsal striatum have intact learning of an escape strategy, reversal-learning, and strategy-shifting [71], whereas infusion of a D2 receptor antagonist in the rat dorsolateral striatum was shown to interfere with the reversal learning of a positively reinforced instrumental response [31].

### 4.4. Food-Restricted D2 Mice Are Characterized by the Pattern of Brain FosB/DeltaFosB Immunostaining Associated with Behavioral Sensitization to Addictive Drugs and Expression of Perseverant and Inflexible Behaviors

The analysis of FosB/DeltaFosB immunostaining revealed strain-specific patterns of expression fostered by restricted feeding in D2 and B6 mice. Thus, whereas food-restricted mice from both inbred strains showed increased FosB/DeltaFosB immunostaining in the mesencephalon, NF B6 mice also showed a strain-specific overexpression of the transcription factor in the BaA and DLS, and NF D2 mice showed a strain-specific overexpression in the CA1 of the hippocampus and in the NAc Sh.

Accumulation of DeltaFosB within the mesencephalon is fostered by prolonged exposure to abuse of drugs and it has been associated with reduced inhibitory control over the activity of mesencephalic DA neurons [72,73,74,75,76]. In D2 mice, a history of either methamphetamine self-administration or of restricted feeding fosters behavioral sensitization to cocaine and also enhanced cocaine-induced firing of mesencephalic DA neurons [23,77], supporting strong similarities between neuroplasticity associated with stress- and psychostimulants-induced behavioral sensitization. On the other hand, behavioral sensitization to amphetamine was observed in food-restricted D2 but not B6 mice [78], whereas mice of both strains developed amphetamine-induced behavioral sensitization [41]. Moreover, repeated exposure to prolonged inescapable and unavoidable stress was reported to reduce responsivity of mesencephalic DA neurons to a natural reward in B6 mice [79]. Therefore, although restricted feeding fosters accumulation of FosB and ΔFosB within the mesencephalon of mice from both strains, this phenotype could be associated with different strain-specific neuroadaptation.

Increased FosB/DeltaFosB immunoreactivity was also observed bilaterally in the SNr of NF B6 mice and in the left SNr of NF D2 mice. The SNr gates striatal output to the thalamocortical part of the cortico-striatal thalamocortical circuits, allowing for both the limbic and motor striatum to influence action through the primary motor cortex [80]. A direct striatal input regulated by the DA receptors of the D1 type inhibits SNr activation and releases motor responses, whereas an indirect striatal input regulated by D2 receptors excites SNr and terminates motor responses. Imbalanced control of the direct and indirect pathways on SNr can foster either stereotyped behavior or apathetic state in primates, depending on the direction of the imbalance [81]. Thus, although the expression of ΔFosB activates a chain of molecular events capable of regulating the excitability of the targeted neurons [82,83,84], a reasonable response to an imbalance between the excitatory and inhibitory striatal inputs, it could result in different behavioral outcomes in the two strains. The lateralized (left) effect of restricted feeding on FosB/DeltaFosB immunoreactivity in D2 mice supports a reduced influence of the indirect pathway to the SNr. The previously discussed reduction in striatal D2 receptors fostered by restricted feeding in mice of this inbred strain is limited to the left hemisphere, and a blockade of D2 receptors in the left dorsolateral striatum is sufficient to reproduce the effects of restricted feeding on stress coping in FF mice of this inbred strain [42]. Finally, a recent study reported that optogenetic inhibition of striatal output neurons of the indirect pathway fosters perseveration in a reversal learning paradigm [85], suggesting a causal relationship between the neural and behavioral phenotypes fostered by restricted feeding in mice of the D2 strain.

A strain-specific increase in FosB/DeltaFosB immunoreactivity was observed in the BaA and DLS of FR mice from the B6 strain. Accumulation of DeltaFosB in the basolateral amygdala was observed as a result of peripheral nerve injury in rats and associated with the emotional and affective component of pain [86], whereas chronic voluntary consumption of large quantities of ethanol induces FosB/DeltaFosB expression selectively in the DLS but not the DMS of rats, an effect mediated by endogenous opioids [87]. As the behaviors we measured were not affected by restricted feeding in mice of this inbred strain, it would be tempting to conclude that the accumulation of DeltaFosB in the BaA and DLS protects against dysfunctional neuroplasticity fostered by the chronic stress.

The strain-specific pattern of brain FosB/DeltaFosB immunostaining observed in NF D2 mice indicates an addiction-like aberrant neuroadaptation to the chronic stressful experience, which could well-support the development of perseverant and inflexible active coping strategies. Thus, NF D2 mice developed a strain-specific increase in FosB/DeltaFosB immunostaining in the CA1 of the hippocampus. These finding are in line with previous observations that: (1) chronic mild stress (CMS)-exposed D2 female mice show a larger hippocampal transcriptional response to an acute stressor than CMS-exposed B6 females [88]; and (2) the CA1 area of the hippocampus shows the largest effect of amphetamine sensitization on gene expression in male D2 mice [89]. Finally, chronic exposure to addictive drugs increases DeltaFosB in the rat hippocampus [90], and both exposure to addictive drugs and viral-mediated overexpression of hippocampal DeltaFosB impairs hippocampus-dependent learning and memory [91] and decrease the excitability of CA1 neurons [92].

Only the NF mice of the D2 strain showed increased FosB/FosB immunoreactivity in the NAc Sh. Accumulation of DeltaFosB in the NAc Sh is observable following repeated exposure or self-administration of addictive drugs in rats and in mice of the B6 strain [93,94], and the accumulation of DeltaFosB in the NAc Sh was found to be involved in stress-induced behavioral sensitization to psychostimulants in rats [34], in line with the previously discussed finding that food-restricted D2 mice strain-specifically develop behavioral sensitization to amphetamine [78]. The accumulation of DeltaFosB in the striatum by prolonged exposure to addictive drugs prevents c-fos induction by an acute drug challenge [95], and we previously reported a strain-specific blockade c-fos induction by acute stress challenge in the NAc Sh of NF+48 D2 mice [40]. Finally, an accumulation of DeltaFosB in the NAc Sh was also observable in rats following a feeding protocol that promotes food addiction [96], and overexpression of FosB/DeltaFosB immunoreactivity in the NAc of rats exposed to palatable food was shown to correlate with binge eating [96]. These findings support a role of DeltaFosB in the NAc in aberrant neuroplasticity associated with compulsive behavior beyond addiction.

Notably, the experience of chronic defeat by a male conspecific accumulates DeltaFosB in both the NAc Sh and core of B6 mice that develop social avoidance [97], and the number of FosB/DeltaFosB immunoreactive cells in the NAc Sh, but not in the dorsal striatum, was found to positively correlate with the severity of stereotypic behavior expressed by female B6 mice living in standard housing conditions [98]. Finally, as discussed, B6 male mice exposed to restricted feeding show perseverant passive coping, suggesting depressive-like disturbances [40,99]. These observations suggest caution in concluding that the pattern of DeltaFosB expression observed in NF mice of the B6 strain indicates resilient adaptation of the chronic stressful experience. Indeed, these data rather support the view that experiences of adverse environments in adult life can foster different dysfunctional adaptations depending on the experience as well as individual sex and genotype.

In conclusion, the pattern of FosB/DeltaFosB immunoreactivity fostered by restricted feeding in the brain of D2 mice identifies plasticity within circuits involved in compulsive, repetitive, inflexible, and perseverant behavioral phenotypes.

### 4.5. Limitations

It is important to point out the limitations of our study, the most relevant being the lack of experiments testing a direct relationship between the behavioral and the neural effects of restricted feeding in mice of the D2 strain. However, the aim of this study was to demonstrate the development of a dysfunctional active coping style through diathesis-stress in a rodent model. Further studies will test the hypothesis that altered processing within the cortico-striatal thalamic cortical circuits, possibly mediated by reduced availability of DA receptors of the D2 type in the dorsal striatum, is involved in the development and expression of dysfunctional active coping by stressed mice of the D2 strain.

## 5. Conclusions

The findings of the present study support the hypothesis that food-restricted mice of the D2 inbred strain develop perseverant/inflexible active coping strategies and brain neuroplasticity associated with drug and food addiction, behavioral sensitization to addictive drugs, and inflexible and perseverant behavior, strain-specifically.

As mice of this inbred strain, when exposed to inescapable and unavoidable stressors, are characterized by an active coping style that is supported by a genotype-specific brain circuitry [12], the present results suggest that an active coping style represents an endophenotype of specific mental disturbances. Endophenotypes are behavioral, cognitive, and biological phenotypes that are intermediates between genetic liability factors and the expression of a given disorder; for this reason, they are not pathological and should be observed in a proband’s asymptomatic relatives [100,101,102]. Moreover, they cut across traditional disorder categories and can be related to etiological processes contributing to comorbidity between different behavioral disturbances [102]. Thus, the observation that an active coping style becomes dysfunctional following the experience of a prolonged life-threatening inescapable and uncontrollable stressor supports its value as endophenotype of disturbances associated with behavioral perseveration, inflexibility, and compulsivity.

## Figures and Tables

**Figure 1 behavsci-11-00174-f001:**
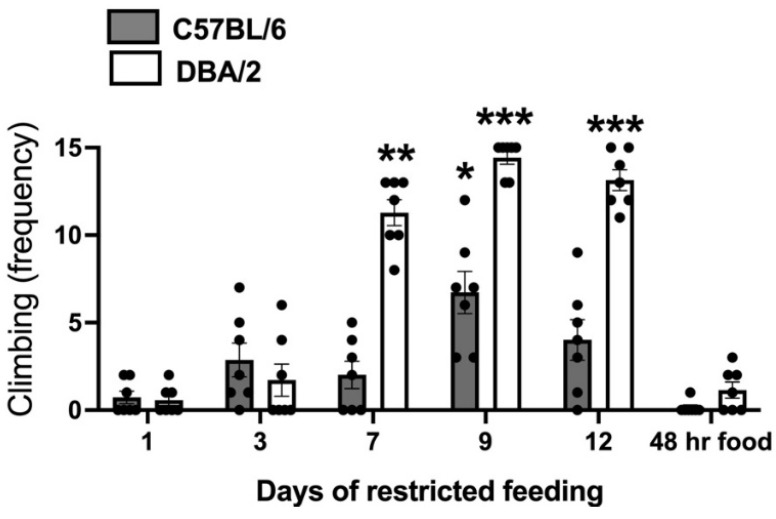
Climbing scores collected in food restricted mice of the C57BL/6 (B6) and DBA/2 (D2) strains in over the 14 days of differential housing. Data are expressed as mean frequencies (± SEM). *; **; *** = *p* < 0.05; 0.01; 0.001 vs. day 1 of differential housing (post-hoc Tukey correction).

**Figure 2 behavsci-11-00174-f002:**
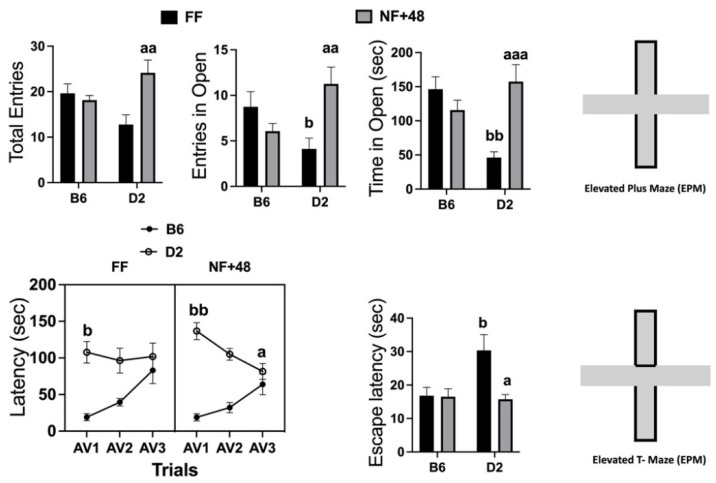
Effects of previous restricted feeding on behavior expressed in the EPM and ETM by C57BL/6 (B6) and DBA/2 (D2) mice. Top: Total entries (open and closed) in the arms of the EPM; number of entries in the open arms of the EPM; time spent in open arms (seconds) of the EPM. Bottom: Latency to emerge from the closed arms of the ETM over the three consecutive experiences (AV1, AV2, and AV3) and latency to escape from the open arms when placed directly into them. Results are expressed as mean ± SEM). a, aa, and aaa = *p* < 0.05, 0.01, 0.001 vs. FF mice of the same strain (post hoc Tukey’s correction). b and bb = *p* < 0.05 and 0.01, respectively, vs. B6 mice of the same group (post hoc Tukey’s correction).

**Figure 3 behavsci-11-00174-f003:**
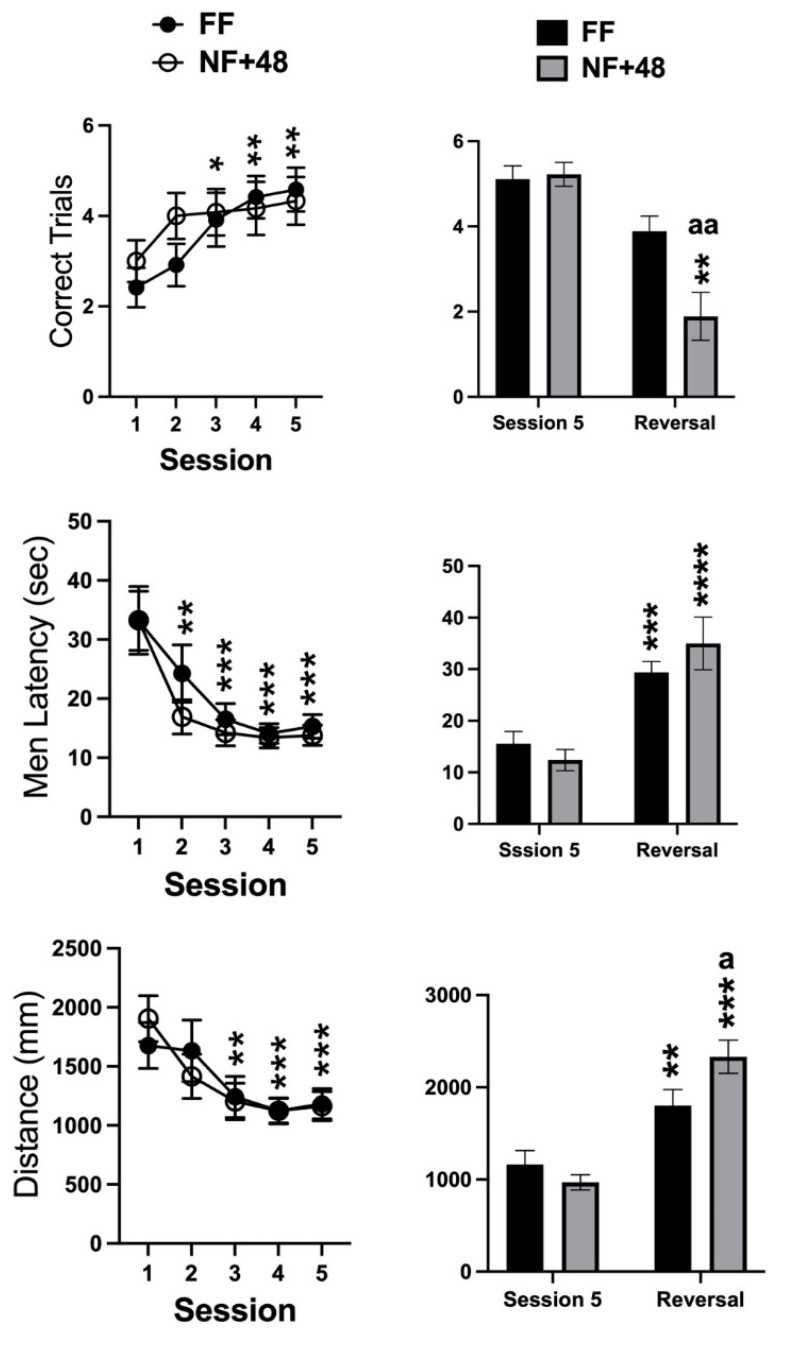
Effects of previous restricted feeding on acquisition and reversal of an escape response by D2 mice. Graphs on the left side show data collected from training sessions of all tested mice; graphs on the right side show data from the fifth and reversal sessions of mice selected for their performance on the last three training sessions. Data are presented as mean (±SEM). *, **, ***, and **** = *p* < 0.05, 0.01, 0.0005, and 0.0001, respectively (post hoc Tukey’s correction) vs. session 1 or session 5. Tukey’s multiple comparisons test. a, aa = *p* < 0.05, 0.01 vs. FF mice. Šídák’s multiple comparisons test.

**Figure 4 behavsci-11-00174-f004:**
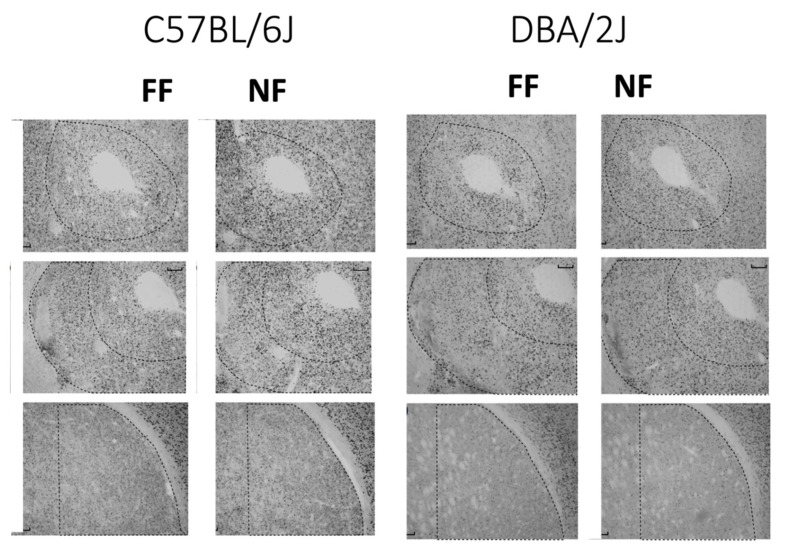
Examples of immunostained tissue. Top: Accumbens Core; middle: Accumbens Shell; bottom: Dorsolateral Striatum.

**Figure 5 behavsci-11-00174-f005:**
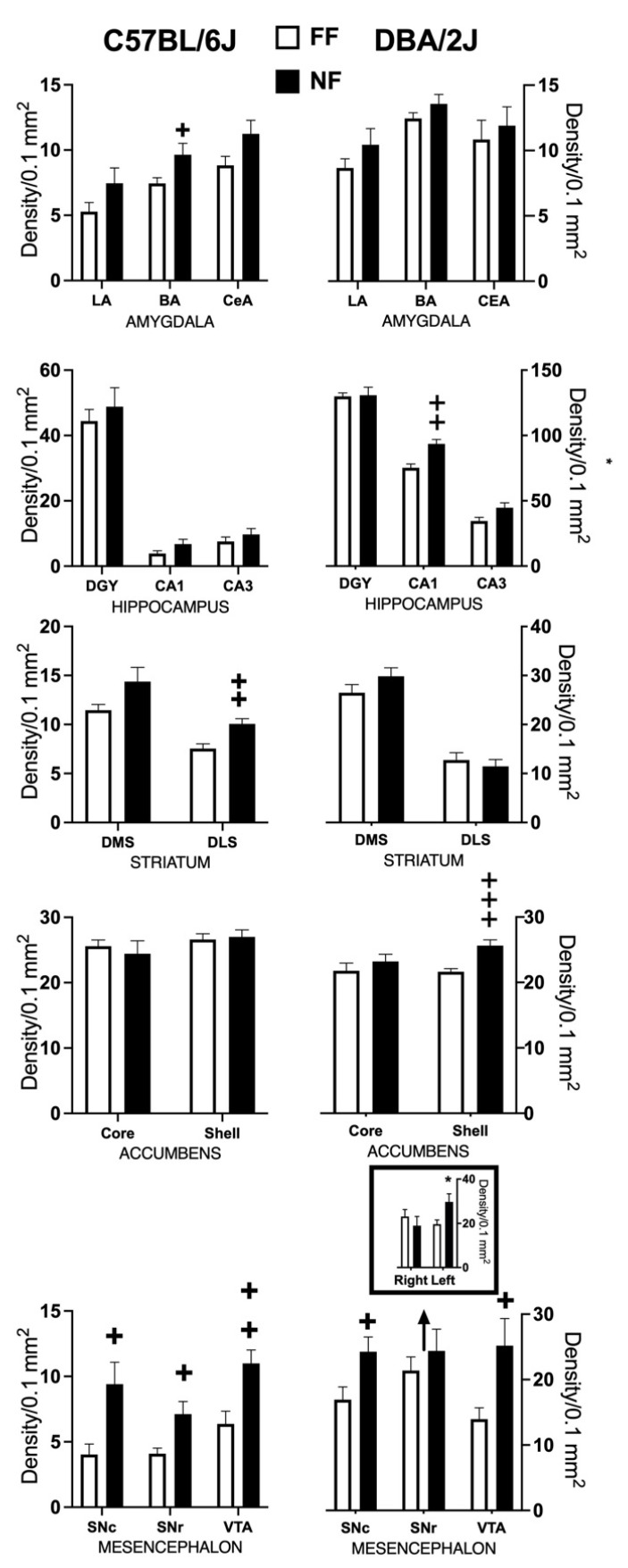
Effects of restricted feeding on FosB/DeltaFosB expression in different brain areas of mice of the DBA/2J and C57BL/6J inbred strains; data are expressed as mean density of immunostained nuclei (±SEM). +, ++, +++ = *p* < 0.05, 0.001, 0.0001 vs. FF (*t*-test, two tailed). * = *p* < 0.05 vs. FF (Fisher’s LSD).

**Table 1 behavsci-11-00174-t001:** Data on the home cage behavior expressed in the 30 min preced-ing food delivery on the 12th day of differential housing.

	DBA/2		C57BL/6	
	FF	NF	FF	NF
STILL	10.8 ± 0.8	0.0 ± 0.0 aa	10.0 ± 1.0	3.8 ± 0.8 a
CLIMB	0.5 ± 0.2	14.2 ± 0.4 §	0.0 ± 0.0	4.0 ± 1.5
REAR	0.2 ± 0.1	0.8 ± 0.3	0.1 ± 0.1	0.0 ± 0.0
LOC	0.2 ± 0.2	0.1 ± 0.1	0.7 ± 0.5	1.1 ± 0.4
DIG	2.2 ± 0.8	0.0 ± 0.0	1.6 ± 0.3	1.3 ± 0.3
GROOM	1.1 ± 0.4	0.0 ± 0.0	2.7 ± 0.6	2.8 ± 0.7
CHEW	0.0 ± 0.0	0.0 ± 0.0	0.0 ± 0.0	0.0 ± 0.0
DRINK	0.0 ± 0.0	0.0 ± 0.0	0.0 ± 0.0	0.0 ± 0.0

a, aa, = *p* < 0.05, 0.01 vs. FF of the same strain. § = *p* < 0.001 vs. all others (Tukey multiple comparisons).

## Data Availability

All data in the manuscript are available upon reasonable request.

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
