# Peer review of "Repetitive and Inflexible Active Coping and Addiction-like Neuroplasticity in Stressed Mice of a Helplessness–Resistant Inbred Strain"

_behavsci, 2021, doi:10.3390/bs11120174_

Round 1

Reviewer 1 Report

Please clarify if the trained observer made use of videos to provide frequency of behaviors? Was the frequency assessed by more than one observer?

Please provide individual scores of behavioral assessments.

Author Response

We wish to thank both reviewers for the helpful suggestions.

  • The paper underwent English editing
  • Introduction, Methods, Results and Discussion were thoroughly corrected according to indications by reviewer 1.
  • These procedures resulted in a very large number of changes that could not be traced without making the MS impossible to read. Therefore, we decided not to show all the revisions on the new MS.
  • Some of the comments of Rev. 1 required discussion and are reported below

Figures and Tables

Most of the figures were modified, some graphic presentations were eliminated: (figure 1)  a table 1 substituted graphs; (figure 2) strain differences were included in the statistical analysis (3 way) and data presented within a single graph; (figure 3) a different set of data was presented.

Figure 4, show samples of the immunostaining of relevant areas so that readers can check the quality of the preparation used to extract quantitative data as well as the correct identification of these areas in the brain of mice from different genetic backgrounds. It is a requirement for this type of experiments and was cited in the result section as well as in the discussion.  We improved the citation in the new version of the manuscript. 

The legends were corrected to offer more information.

The ‘mini graph’ in Figure 5 was cited in the result section of the immunohistochemical experiment. In the new version of this section the reason of presenting this finding should be clearer. Moreover, these data were discussed because they were a relevant support to the proposed functional circuit associated with dysfunctional active coping.

Q: Lines 349-354 The argument presented here does not seem to support the idea that a stereotypy has developed as a result of the food deprivation condition. If the emergence of cage climbing is directly linked to food searching behaviour, and thus the motivation to obtain food, (and not foodacquisition) then it cannot be classified as stereotypy because it clearly has a function (i.e. searching for food), regardless of whether food ispresent or not. Searching for food is no guarantee of acquiring food and thus whether the behaviour results in feeding has no bearing on theclassification of the behaviour as searching. The behaviour may be repetitive or even habit forming but it does not necessarily constitute stereotypy (the authors do not provide a definition for the term and so

it is difficult to determine whether they are considering stereotypy as it

is typically defined in the literature). Furthermore, the argument

presented here is directly tied to motivation to find food, which the

preceding statement implies is outside of the explanatory realm for the

observed patterns in behaviour.

R: We respect the opinion of the reviewer, but we stick to our own. We indeed presented a definition of stereotypy in the first version of this paper: " ...an inflexible/perseverant behavior with no apparent function that is observable in animals exposed to sub-optimal environmental conditions", followed by an appropriate reference. There are 2 relevant elements in this definition: 1) no apparent function, 2) observable in animal exposed to sub optimal conditions. In the new version we added the reported evidence that stereotypic behavior expressed by caged animals can be the outcome of ‘frustrated motivation’ to further support our conclusion.

Q: Lines 365 The use of the terms 'safe' and 'dangerous' does not seem appropriate

here. Perhaps 'aversive' and 'non-aversive' would be better terms to use,

given that they do not assume anything about the nature of the spaces

in the maze tests and relate only to the perception of the test animal

itself. After all, neither of the sections in the mazes were actually any

more safe or dangerous than the other but may have been perceived as

such by the mice.

R: We do not understand the point, we are measuring behavior expressed by animals experiencing a stressful situation. For both human and non-human animals a stressful situation is a condition perceived as such. A borrow is a safe place for rodents while an open space is not and this difference influences their behavior. Nonetheless, since the phrasing and the language used in the first version of the paper could have been misleading we rephrased it.

Q: Line 389 What is meant by the term "extreme freezing"? The term is ambiguous.

Was it cold in the room? Are they referring to immobility? What

makes the freezing response "extreme"? Furthermore, what do the

authors attribute this to? Is it a clear fear response? Up to this point the

authors have been discussing passive avoidance of the aversive

stimulus of the open arms of the maze but this seems far more like a

fear response. Why would FF D2 mice experience the open arms as

eliciting fear to the extent of immobility, rather than simply not

entering the open arms, whereas FF B6 mice do not? Moreover, why

would this fear response be present from the first introduction into the

maze (i.e. without prior maze experience)?

R: the term ‘extreme’ was substituted with ‘high levels of’. The definition of ‘freezing’ as an ambiguous term is somewhat surprising. The term is used in behavioral sciences to describe a state of rigid immobility that is elicited by an impending danger (such as potential predation). The lack of movements reduces the chances to be detected. The response can be conditioned (fear conditioning) but it does not require the immediate presence of a danger or of fear-associated conditioned stimuli. High levels of freezing responses have been shown to interfere with learning of active avoidance early during training and, of course, they prevent exploration of a novel environment as well as the search of a way to escape aversive contexts. For this reason, freezing is considered a passive defensive/coping response.  Therefore, the behavior expressed by D2 mice is not surprising but it is ‘extreme’ because it is expressed in to a level that interferes with successful adaptation to the novel environment: i.e. initial exploration followed by development of a passive avoidance of areas exposed to species-typical predation. In the new version we clarified these points.

Q: Lines 389-390 It seems like the FF d2 mice had an innate avoidance of the open arms

such that there was no need to develop an avoidance thereof. To claim

an interference effect suggests that there was a need to develop the

avoidance of the open arms, which does not seem to be the case based

on the description provided here.

R: All rodents show innate avoidance of the open arms in an elevated maze. This is what is exploited by these standard anxiety tests. However, even innate avoidance requires the knowledge of the areas that should be avoided, and this requires initial exploration of the novel context.  Rats and B6 mice initially leave the protected arms to explore the novel environment and, through this behavior, acquire the knowledge that the environment includes a protected (safe) and unprotected (dangerous) area. It is the progressive knowledge of the unprotected arms that supports progressive development of a passive avoidance. We think that the new version of the paper guides the reader through the reasoning beyond the protocols and measures used to experimentally test emotional behavior in rodents.

Q: Line 399 Do the authors mean to say the latency to leave the protected arm?

Using the word "escape" implies that there is some kind of threat

present in the protected arm, which in the context of the experiment

does not seem to make sense.

R: Escape and freezing are innate reactive responses associated with high levels of negative emotion in human and non-human animals. Either freezing or escape are behavioral measures of ‘fear’ in animal models. As freezing is a passive defensive coping, escape is the prototype of active defensive coping. FF D2 mice freeze within the protected arm because of fear of the unknown environment they have been forced in; NF+48 D2 mice dart out of the same area for the same reason. We tried to clarify these points in the new versions.

Q: This conclusion seems unfounded. The described responses of the mice

sound as though the NF+48 mice were merely more active in the maze

than the FF mice were. To conclude that this is evidence of a "loss of

control over reactive escape responses" seems to ignore possible

alternative explanations. Perhaps the mice are active because they have

been starved for an extended period and are behaviourally primed to

engage in searching behaviour. Were these mice more active in their

home cages relative to the FF mice? Is there a possible physiological

(not necessarily neurological) process causing more active behaviour

in the maze?

R: High levels of activity and searching explain exploration, not escape. FF D2 mice were not less active than NF+48 mice of the same strain; they froze in the open arms as they did in the protected one. NF+48 mice showed less freezing not more active exploration. Their active behavior could be described as darting, unfortunately we did not quantify it. This point was clarified in the new version of the paper

Q: Line 406 Why would extreme reactive escape responses be maladaptive? Surely

escape from an aversive stimulus would be an adaptive response,

regardless of the extent or intensity of the reaction. How would escape

responses, regardless of type, enhance risk?

R: We understand that this part could have been confusing for readers who are not specifically expert of experimental methods used by animal research on emotional behavior, we detailed it.

Q: Line 419 What do the authors consider to be characteristic of a "perseverative"

response in this context? The mice were only tested once on the

reversal learning task, so to classify any response as perseverative in a

context where there is only one observation seems unfounded. This is

important given that the authors claim that their study offers "strong

support to the hypothesis that previously food restricted 426 D2 mice

develop inflexible/perseverant active coping in novel stressful

situations".

R: In the first version of the manuscript we indicated a specific reference for this topic in the new version we specified the motivation presented in the cited paper (1553-1558).

Q: Lines 494-495 Was this despite the accumulation of ΔFosB? How do these

observations relate to FosB expression? Simply observing that a strain

displays a behaviour and also displays a characteristic FosB expression

does not mean that they are related. The authors should explain the

logic and the connection here more explicitly.

R: We moved these considerations at the end of the discussion where the context could facilitate the understanding. Nonetheless it is worth pointing out that FosB findings were presented and discussed in the typical way used to present and discuss biological data whose causal relationship with behavioral measures was not tested. In the latter case the term “demonstrate” would have been used and indications of the direction of the causal relationship would have been detailed. The absence of a causal demonstration was explicitly acknowledged and the motivation of the interest of the data was given in the final part of the discussion.

Reviewer 2 Report

The presented study seeks to demonstrate that D2 inbred mice develop dysfunctional active coping strategies following a period of food deprivation. This is done by subjecting said mice to a period of food deprivation and administering subsequent tests of anxiety in addition to examination of brain neuro-anatomical changes. There are two major issues with the submission as it is.

The first issue is one of language and communication, and relates directly to the second major issue. The paper is very difficult to read because the language use is unnecessarily complicated and, at times, inappropriate. It is evident that the submission should have undergone further proof-reading an editing prior to being submitted. While English may not be the first language of the authors, it is still standard practice to have the manuscript proof-read prior to submission. Many minor mistakes complicated the interpretation of what was being communicated which could easily have been avoided.

While this may seem pedantic, it is important for two reasons. Firstly, the journal is an open-source publication and thus is likely to reach a far wider audience relative to pay-per-view journals. As such, it is advisable that the authors make as much effort as possible to communicate their findings in a way that would be easily understood by most people. Secondly, and more importantly, the manner in which the paper is written makes interpretation of the science very difficult.

The latter constitutes the second major issue with the manuscript. It is very difficult to interpret the study methods and findings because they are communicated in a manner which is obscure and confusing. Many of the terms used are ambiguous or have a pre-existing meaning in the literature which the authors do not appear to be adhering to in their use and interpretation of the terms. As such many of the conclusions drawn from the data seem unfounded and the logic is not clear. The manner in which the data are presented further complicates interpretations. At many points it is clear that the authors have a sense of their own data and interpretations but leave it to the reader to make the connections themselves.

It is my opinion that the basic science is not compromised and the study has the potential to provide a valuable datum which may have important consequences for many fields of study. However, unless the authors are able to improve the presentation of their study, it will be of little value. For this reason I am recommending that the study be given further consideration following major revision.

Author Response

(The authors gave the same response as above.)

Round 2

Reviewer 2 Report

I am content with the changes that the authors have made and appreciate the effort which they have clearly put into improving the submission. The manuscript is much clearer now and there is far less ambiguity in the language use and terminology. There are a few points which I wish to address relating specifically to the comments of the authors on my feedback to them around the issue of stereotypy. For the sake of clarity, I quote my original feedback to the authors below as well as their response. I then present my response in italics.

“Q: Lines 349-354 The argument presented here does not seem to support the idea that a stereotypy has developed as a result of the food deprivation condition. If the emergence of cage climbing is directly linked to food searching behaviour, and thus the motivation is to obtain food (and not food acquisition), then it cannot be classified as stereotypy because it clearly has a function (i.e. searching for food), regardless of whether food is present or not. Searching for food is no guarantee of acquiring food and thus whether the behaviour results in feeding has no bearing on the classification of the behaviour as searching. The behaviour may be repetitive or even habit forming but it does not necessarily constitute stereotypy (the authors do not provide a definition for the term and so it is difficult to determine whether they are considering stereotypy as it is typically defined in the literature). Furthermore, the argument presented here is directly tied to motivation to find food, which the preceding statement implies is outside of the explanatory realm for the observed patterns in behaviour.

R: We respect the opinion of the reviewer, but we stick to our own. We indeed presented a definition of stereotypy in the first version of this paper: " ...an inflexible/perseverant behavior with no apparent function that is observable in animals exposed to sub-optimal environmental conditions", followed by an appropriate reference. There are 2 relevant elements in this definition: 1) no apparent function, 2) observable in animal exposed to sub optimal conditions. In the new version we added the reported evidence that stereotypic behavior expressed by caged animals can be the outcome of ‘frustrated motivation’ to further support our conclusion.”

Firstly, I would like to apologize – the authors do indeed present a definition for stereotypy which I overlooked during the review of their submission. However, I would still argue that the interpretation of the described behaviour as stereotypy without supporting evidence based on their observations (e.g. descriptions of frequency, duration or repetitive form) is not sound. Firstly, the description of the behaviour 'CLIMB' reads, "climbing (CLIMB), clinging with four paws to the cage cover"; this does not specifically include any criteria which would identify the behaviour as a stereotypy (e.g. repeated for periods in excess of 30sec, or, occurring more than 3 times in succession). When reporting in the results, the authors do not report on these characteristics either but only report on an increased frequency of observation of CLIMB behaviour; observing an increase in the frequency of a behaviour alone does not constitute stereotypy. Secondly, the definition provided specifically requires that the behaviour appear functionless. The problem for the authors lies in the fact that the behaviour is directed toward the area of the cage associated with feeding and food under normal circumstances in a condition where the animals themselves are starved for an extended period of time. The animals have been conditioned (either explicitly or implicitly through repeated feeding in the same area of the cage prior to their involvement in the experiment) to associate the roof of the cage and the cage hopper with food. Thus it is not possible to know with any degree of certainty that activity directed to this area of the cage in a starved animal is devoid of a food acquisition function. The authors cannot reasonably show that the behaviour is NOT ineffective searching for food in a hungry animal and thus, it is not possible in this context to determine whether the behaviour is truly functionless. If the behaviour were present in a condition where food was freely available and there was evidence that the behaviour occurred in a manner which is outside the norm (e.g. repeated consecutively more than 3 times) then I would agree wholeheartedly with the authors in identifying the behaviour as stereotypic. Were the authors discussing an alternative form of behaviour (such as looping or somersaulting) where a food-centric motivation was absent, then the label of stereotypy would be appropriate. However these are not the case. It seems more reasonable, given the absence of quantitative evidence to justify labeling the behaviour as stereotypic, to err on the side of caution and simply label the behaviour as increased cage climbing. The fact that the authors also report that climbing behaviour returned to baseline levels once food was provided suggests that the behaviour was not stereotypic; as described above, if the behaviour persisted in the presence of food, then it could be considered stereotypic.

There is also the fact that the authors describe the feeding of NF mice as being on the floor of the cage – why are the NF mice being fed at all? This treatment group was specifically described as being starved for 12 days prior to testing (hence the label ‘NF’ for ‘No Food’). This requires clarification. Are the authors referring to after the experiment (in which case it has no bearing on the classification of the behaviour as stereotypic under an entirely different set of circumstances)?

Author Response

Rev: Firstly, I would like to apologize – the authors do indeed present a definition for stereotypy which I overlooked during the review of their submission. However, I would still argue that the interpretation of the described behaviour as stereotypy without supporting evidence based on their observations (e.g. descriptions of frequency, duration or repetitive form) is not sound. Firstly, the description of the behaviour 'CLIMB' reads, "climbing (CLIMB), clinging with four paws to the cage cover"; this does not specifically include any criteria which would identify the behaviour as a stereotypy (e.g. repeated for periods in excess of 30sec, or, occurring more than 3 times in succession). When reporting in the results, the authors do not report on these characteristics either but only report on an increased frequency of observation of CLIMB behaviour; observing an increase in the frequency of a behaviour alone does not constitute stereotypy. Secondly, the definition provided specifically requires that the behaviour appear functionless. The problem for the authors lies in the fact that the behaviour is directed toward the area of the cage associated with feeding and food under normal circumstances in a condition where the animals themselves are starved for an extended period of time. The animals have been conditioned (either explicitly or implicitly through repeated feeding in the same area of the cage prior to their involvement in the experiment) to associate the roof of the cage and the cage hopper with food. Thus it is not possible to know with any degree of certainty that activity directed to this area of the cage in a starved animal is devoid of a food acquisition function. The authors cannot reasonably show that the behaviour is NOT ineffective searching for food in a hungry animal and thus, it is not possible in this context to determine whether the behaviour is truly functionless. If the behaviour were present in a condition where food was freely available and there was evidence that the behaviour occurred in a manner which is outside the norm (e.g. repeated consecutively more than 3 times) then I would agree wholeheartedly with the authors in identifying the behaviour as stereotypic. Were the authors discussing an alternative form of behaviour (such as looping or somersaulting) where a food-centric motivation was absent, then the label of stereotypy would be appropriate. However these are not the case. It seems more reasonable, given the absence of quantitative evidence to justify labeling the behaviour as stereotypic, to err on the side of caution and simply label the behaviour as increased cage climbing. The fact that the authors also report that climbing behaviour returned to baseline levels once food was provided suggests that the behaviour was not stereotypic; as described above, if the behaviour persisted in the presence of food, then it could be considered stereotypic.

Authors: We fully described the behavior expressed by D2 mice in the Discussion: Lines 726-731 in the new version

Reviewer: cage – why are the NF mice being fed at all? This treatment group was specifically described as being starved for 12 days prior to testing (hence the label ‘NF’ for ‘No Food’). This requires clarification. Are the authors referring to after the experiment (in which case it has no bearing on the classification of the behaviour as stereotypic under an entirely different set of circumstances)?

Authors: We added the missing description of the scheduled feeding protocol in the Methods: lines 164-167